# Beyond GRACE: Evaluating the benefits of NGGM and MAGIC for precipitation estimation over Europe

Muhammad Usman Liaqat<sup>1</sup>, Stefania Camici<sup>1</sup>, Francesco Leopardi<sup>1,2</sup>, Jaime Gaona<sup>1</sup>, Luca Brocca<sup>1</sup>

<sup>1</sup>Research Institute for Geo-Hydrological Protection, National Research Council of Italy, Via Madonna Alta, 126, 06128 Perugia Italy.

<sup>2</sup>Department of Civil and Environmental Engineering, University of Perugia, Italy.

Correspondence: muhammadusman.liaqat@cnr.it

#### Abstract

20

25

30

The Gravity Recovery and Climate Experiment (GRACE) mission and its Follow-On (GRACE-FO) mission provide observations of terrestrial water storage (TWS) dynamics from regional to global scales. However, they lack high spatio-temporal resolution, which is essential for hydrological applications. A join collaboration between the National Aeronautics and Space Administration (NASA) and the European Space Agency (ESA), initiated a decade ago, is known as the Mass change And Geosciences International Constellation (MAGIC). The aim of this collaboration is to launch a new paired mission, i.e., GRACE-C and NGGM (Next Generation Gravity Mission), to improve the monitoring of extreme events such as floods and droughts.

The primary objective of this study is to examine the impact of the expected improvement in the spatial-temporal resolution and accuracy of NGGM and MAGIC on precipitation estimation by developing multiple synthetic experiments on a European scale. The study employed the well-known SM2RAIN algorithm to estimate the precipitation accumulated between two consecutive TWS measurements. The total amount of water in the soil from the fifth generation ECMWF reanalysis for the land (ERA5L) is used as a proxy of TWS for the period of 2003-2012. Firstly, the reliability of SM2RAIN to obtain precipitation from TWS measurements is tested by using ERA5L precipitation as reference. The results showed that SM2RAIN exhibited satisfactory performance at a daily temporal resolution, with mean values of the correlation coefficient, R, equal to 0.86. Good agreement was obtained across most of Europe except in some areas of the northern Italy, northeastern states (Estonia, Latvia) and coastal regions. Secondly, synthetic experiments 'were developed by degrading the temporal resolution of TWS data and by introducing error ranging from 1 to 40 mm. The results showed that while the SM2RAIN algorithm maintains robust performance under moderate temporal degradation (5-10 days) and low measurement errors (





#### 1. Introduction

Precipitation is one of the primary components of the hydrological cycle, the precise estimation of spatio-temporal distribution of precipitation plays a vital role in assessing extreme events (Vrochidou et al., 2013), water resources management (Dorigo et al., 2021) ecological assessments (Yu et al., 2024) and in hydrological research (McMillan et al., 2011; Meles et al., 2024).

Primary methods to estimate precipitation include ground observations (rain gauges and radar), reanalysis datasets and satellite based products (Song et al., 2020; Yuan et al., 2020). Although rain gauges are marked as one of the reliable sources of precipitation measurement, their spatial coverage, especially in mountainous and data-scarce regions, remains a primary limitation for their applicability on a global scale (Ahrens, 2006; Herrnegger et al., 2018; Kidd et al., 2017). Meteorological radar provides high-resolution spatial estimates of precipitation. However, it is constrained by limited accessibility and high operational costs (Sun et al., 2018). Reanalysis datasets such as the 5<sup>th</sup> reanalysis from the European Centre for Medium-Range Weather Forecasting (ECMWF), ERA5, have been developed and widely used from regional to global scales. Nevertheless, the accuracy of the reanalysis product depends upon the multi-source data, including in-situ observations and satellite products, which constrains its applicability in a data scarce mountainous environment (Ansari et al., 2022; Li et al., 2020; Liaqat et al., 2022). Satellite based precipitation products with their high spatial and temporal coverage, provide an alternative solution for precipitation estimation.

There are two primary methods for precipitation estimation from satellite, including top-down and bottom-up approaches (Brocca et al., 2014). In the top-down approach, precipitation is based on the emitted or reflected radiation from clouds or rain droplets derived from a combination of Geostationary (GEO) and Low Earth Orbiting (LEO) satellites sensors (Levizzani et al., 2020). In the bottom-up approach, precipitation is computed by soil moisture (SM) observations inverting the soil water balance in the so-called SM2RAIN algorithm (Brocca et al., 2013, 2014). A key difference between two approaches is that the top-down provides instantaneous measurements of precipitation which can be underestimated when the satellites do not pass over the precipitation event while the bottom-up approach gives accumulated precipitation between two subsequent SM measurements.

SM2RAIN has already been implemented from regional scale in South Asia (Lai et al., 2022) to global scale (Brocca et al., 2019), in Africa (Hengl et al., 2021; Islam, 2022), America (Paredes-Trejo et al., 2019; Satgé et al., 2020), Australia (Chua et al., 2022) and Europe (Moges et al., 2022). Various SM2RAIN products have been developed by forcing different SM satellite observations into the algorithm, including ASCAT (Advance Scatterometer (Brocca et al., 2019), Soil Moisture and Ocean Salinity (SMOS) (Brocca et al., 2016), Advanced Microwave Scanning Radiometer 2 (AMSR2) (Tarpanelli et al., 2017), Sentinel-1 (Filippucci et al., 2022).

In this study, SM is replaced for the first time with total water storage (TWS) to estimate precipitation between two consecutive measurements. The estimation of TWS from predecessor gravity missions, Gravity Recovery and Climate Experiment (GRACE) (Tapley et al., 2004), Gravity Field and steady-state Ocean Circulation (GOCE, Drinkwater et al., 2003) and GRACE Follow-On (Landerer et al., 2020) suffers from low temporal resolution (monthly), making it challenging to accurately estimate precipitation except for areas affected by strong amounts, e.g., tropical areas. Currently, TWS estimated at daily scale is only achievable in conjunction with models (Croteau et al., 2020). Thus, it

https://doi.org/10.5194/egusphere-2025-3659 Preprint. Discussion started: 17 October 2025 © Author(s) 2025. CC BY 4.0 License.







is indispensable to work on high temporal resolution satellite-based solutions from daily to weekly scales to improve our capability to estimate global scale precipitation.

A joint collaboration between the National Aeronautics and Space Administration (NASA) and the European Space Agency (ESA) was initiated a decade ago known as the Mass change And Geosciences International Constellation (MAGIC) (Daras et al., 2023). The ultimate objective of this collaboration is to improve the current models and launch new high-resolution missions to improve capacity for monitoring extreme events such as natural hazards, droughts and floods. The European Next-Generation Gravity Mission (NGGM), a component of MAGIC, is presently undergoing its Phase A Extension as the initial Mission of opportunity under ESA's Future EO Program. Within the framework of international cooperation, ESA and NASA have coordinated studies on gravity constellations to optimize the retrieval of mass change and transport in the Earth system. The new high spatio-temporal resolutions allow for new potential applications and the ability to generate short-term (i.e. fast-track) gravity products on a sub-weekly basis (Daras et al., 2024; Haagmans and Tsaoussi, 2020).

The objectives of this study are: 1) to examine the performance of SM2RAIN algorithm to estimate precipitation using TWS data as input, 2) to investigate the impact of the improved accuracy and spatiotemporal resolution of NGGM and MAGIC on precipitation by using the SM2RAIN algorithm on a European scale, and 3) to analyse the impact of different configurations by adding Gaussian error on simulated precipitation estimates.

## 2. Material and methods

#### 2.1 Study area

The study focuses on the European region, specifically the area with latitudes 30° to 60°N and longitudes 10°W to 50°E (see Fig. 1). This extensive domain is characterized by diverse climatic zones, each playing a critical role in shaping precipitation dynamics and hydrological processes. Specifically, according to the Köppen-Geiger climate classification, the European region exhibits significant climatic diversity: the Oceanic Climate (Cfb), mainly evident in Western Europe along the Atlantic coastline and largely regulated by the proximity to the prevailing westerlies and ocean. It is characterized by mild temperatures and relatively consistent high precipitation throughout the year. In contrast, the Continental Climate (Dfb, Dfc), largely prevails in Central and Eastern Europe where seasonal extremes especially for temperature are more pronounced, characterised by cold winters and warm summers. Precipitation is generally moderate with peak event in summer. The Mediterranean Climate (Csa, Csb) is distinguished by hot dry summers and mild wet winters which largely dominates in Southern Europe and in the Mediterranean Basin. Further, the Polar and Subarctic Climates (Dfc, ET) are experienced in Northern Europe including Scandinavia, and high-altitude regions such as the Alps featured by long harsh winters and short summers. Precipitation is generally low but persistent, with significant amounts of snowfall during the winter, feeding into glacial and snowmelt-driven hydrological processes.




Figure 1: Geographical extent and Climatic classification based on Köppen-Geiger climate types of the Europe. The map on the left is based on OpenStreetMap data © OpenStreetMap contributors (available under the Open Database License).

# 2.2 ERA5L Soil Moisture and Precipitation data

The study utilized the volumetric soil moisture content and the total precipitation available from the European Reanalysis 5<sup>th</sup> Generation ERA5L which is the land component of ERA5, the global reanalysis product developed by ECMWF (Muñoz-Sabater et al., 2021). In particular, the volumetric soil moisture content (m³/m³) at four soil depth layers (i.e., Layer 1, 0–7 cm, Layer 2, 7–28 cm, Layer 3, 28–100 cm, Layer 4, 100–289 cm) were utilized to estimate TWS. ERA5L based TWS is highly correlated with GRACE based TWS as shown in Fig S1, thus it is a reliable proxy for TWS estimation. The TWS from ERA5L is estimated by calculating the product, layer by layer, between the soil moisture content and the layer thickness. The product facilitates the conversion of volumetric moisture content into the depth of water stored within the soil column. The final TWS estimate in mm can be obtained by adding the water stored in all four layers. The equation used to calculate TWS is:

$$TWS = (SM_1 * 0.07) + (SM_2 * 0.21) + (SM_3 * 0.72) + (SM_4 * 1.89) * 1000$$
 (1)

where SM<sub>i</sub> depicts volumetric soil moisture content in (m³/m³) for the i-th layer. The constants (0.07, 0.21, 0.72, and 1.89) are the thicknesses in meters of the corresponding soil layers. The calculation provides the TWS in the soil column in meters of water equivalent. ERA5L total precipitation was considered as benchmark for performance assessment of SM2RAIN algorithm in this study. ERA5L is available at 0.1° and hourly temporal resolution from 1950 to present. In this study, the analysis period was constrained to the timeframe between 2003 and 2012, and the data were aggregated on a daily basis and resampled at 100 km resolution.

#### 2.3 The SM2RAIN algorithm

A modified version of the SM2RAIN algorithm, i.e., using TWS instead of soil moisture, was applied to estimate precipitation between two successive TWS measurements in the time interval *dt*, by inverting soil water balance equation as follows:




$$Z^* \frac{dTWS^*(t)}{dt} = p(t) - r(t) - e(t) - g(t)$$
(2)

Here  $Z^*$  [mm] represents the water capacity of the soil layer, calculated as the product between the depth of the soil layer and its porosity. The term TWS\* refers to the relative TWS [–] (i.e., ranging between 0 and 1), t [T] is the time, p is the precipitation rate [mm T<sup>-1</sup>], r(t) is the surface runoff rate [mm T<sup>-1</sup>], e(t) is the evaporation rate [mm T<sup>-1</sup>] and g(t) is the drainage rate [mm T<sup>-1</sup>]. During precipitation events, evaporation and surface runoff are assumed negligible (Brocca et al., 2015). The drainage rate is dependent on TWS, which is derived from Famiglietti and Wood's (Famiglietti and Wood, 1994). Thus, the precipitation rate can be estimated by simplifying Eq. (2) as follows:

$$p(t) = Z^* \frac{dTWS^*(t)}{dt} + as(t)^b$$
(3)

where a (mm T<sup>-1</sup>) and b (-) refer to the saturated hydraulic conductivity and the exponent of the Famiglietti and Wood equation, respectively. The estimation of parameters a, b, and Z\* is needed to compute the precipitation between two consecutive *TWS*\* measurements. The SM2RAIN application acquires these three parameters by calibrating them against a reference precipitation with comparable spatial and temporal resolution. Prior to the running of the algorithm, the noise in the TWS signal is reduce by applying the exponential filter as described by Brocca et al., (2019), where the characteristic time length parameter (T) is considered a function of SM through a two parameter power law. The value of the two parameters of the modified exponential filter equation, and the a, b, and Z\* parameter values are obtained in this study by point-by-point calibration of SM2RAIN algorithm against the ERA5L precipitation dataset. This calibration involves comparing simulated precipitation with reference data (e.g., ERA5L), where model performance is assessed using the Nash–Sutcliffe efficiency (NS) index.

## 145 2.4 Setup of the synthetic experiments

The study employed the SM2RAIN algorithm to estimate the precipitation accumulated between two consecutive TWS measurements (Brocca et al., 2014). It utilized ERA5L integrated soil moisture for the 4 soil layers as a proxy of TWS, and ERA5L precipitation as a reference. The analysis was carried out for the period 2003-2012 period and it covers the whole European domain. Both TWS and precipitation datasets were spatially gridded into 100 km spatial resolution using the bilinear interpolation method.

As a first step, the ERA5L based TWS at daily time scale was used as input to SM2RAIN and ERA5L precipitation is used as reference. This step was carried out for checking the reliability of SM2RAIN in estimating precipitation from TWS measurements as the approach was previously applied only using soil moisture measurements as input (e.g., Brocca et al., 2014).

In a second step, synthetic experiments were conducted by resampling daily TWS time series to 5-day intervals serving as a proxy of NGGM/MAGIC missions which will provide such temporal resolution. Further, Gaussian errors (1.9 mm, 4.2 mm, 19 mm and 42 mm) were incorporated into the resampled TWS data to assess model performance as a function of the expected error values of the mission. These error values are the expected uncertainty values from the NGGM mission for the short-term time scales and for spatial resolutions of 400 km and 800 km (see Table 11 in (Daras et al., 2023)). As reference, the simulated precipitation from the first step was used. This step was carried out to assess





the performance of four different configurations (5-day sampling, 4 error levels) in estimating precipitation that will resemble the expected configuration available from NGGM mission.

In a third step, the combined impact of temporal aggregation and error level was assessed by considering temporal resolution of 5, 10, 15 and 30 days for TWS measurements, and error levels from 0 mm to 40 mm. The results were evaluated over 100 random locations in Europe. The final step was carried out to investigate the potential improvements of NGGM and MAGIC missions with respect to GRACE\-FO mission.

The obtained results were tested at with a temporal aggregation of 15-day and 30-day, as having TWS measurements every 5 days and differentiating the signal, precipitation can be estimated reliably for a temporal aggregation 3 times the resolution of the input TWS measurements (Brocca et al., 2013). The performance scores used to evaluate the results were: root mean square error (RMSE), coefficient of correlation (R) and BIAS (mm).

#### 3. Results and Discussion

## 3.1 SM2RAIN reliability using TWS measurements as input

The assessment of the SM2RAIN algorithm for estimating 15-day and 30-day precipitation using ERA5L based TWS is presented in Fig.2. The results show that SM2RAIN exhibited satisfactory performance, with mean values of R and RMSE equal to 0.86 and 13.31 mm, respectively, for the 15-day and equal to 0.86, 21.06 mm for the 30-day temporal assessment against ERA5L precipitation. The mean value of BIAS indicated a slightly larger underestimation at 15-days (-0.31 mm) with respect to 30-day (-0.23 mm). Significant negative values of BIAS were found across most of Europe, with the strongest underestimation bias observed in the central and northeastern Europe. Although, the Mediterranean region and southern Europe also exhibits negative bias values, their impact generally less intense than Northen regions. Based on the statistical analysis, SM2RAIN-simulated precipitation shows good agreement across most of Europe, with exceptions in the Alpine areas in northern Italy, in the northeastern states (Estonia, Latvia) and in the coastal regions of Norway. The temporal plot of mean simulated and observed precipitation for 15-day aggregation during 2003-2012 further supports the above results discussed above, as shown in Fig.3. The consistent overlap of both low and high precipitation limbs supports the consistency of the simulations in capturing the observed variability. Overall, these results pave the way for assessing the suitability of SM2RAIN to simulate precipitation for the first time using TWS, and they support the development of synthetic experiments for future gravity missions.



Figure 2: Performances of SM2RAIN algorithm forced with ERA5L based TWS against the ERA5L precipitation data over Europe for the period 2003-2012. The two columns refer to 15-day and 30-day aggregation period., rows indicate the correlation coefficient, R (top panel), BIAS (middle panel) and root mean square error, RMSE (lower panel).

Figure 3: Mean value of precipitation for 15-day aggregation as estimated from TWS through SM2RAIN and obtained from ERA5L precipitation

## 3.2 Performance of NGGM synthetic TWS for precipitation estimation

In this section, synthetic experiments were designed to assess the performance of the NGGM mission. SM2RAIN derived precipitation is used as reference, ERA5L based TWS with 5 day aggregation and introducing an error of 1.9 and 4.2 mm (target error at 800 km and 400 km resolution, Daras et al., 2023) and 19 mm and 42 mm (threshold error at 800 km and 400 km resolution, (Daras et al., 2023) was used as input.







Figure 4: Performance of SM2RAIN algorithm as a proxies of the future NGGM mission forced with ERA5L based TWS at 5 days resolution over Europe for the period 2003-2012 and introducing an error from 0 mm to 42 mm for a 15-day aggregation period. The top panels shows the correlation coefficient, the middle panels the BIAS, and the lower panels the root mean square error, RMSE.

The temporal resampling of data to a 5-day interval generally maintains a good agreement with the reference precipitation data as evident from all statistical measures with R, RMSE, and BIAS values of 0.95, 9.45 mm, and -0.21, respectively (see Fig. 4, No Error plot). Stronger correlation (R values above 0.88) were observed in the southern Europe and parts of Mediterranean regions. A similar pattern was noted for RMSE values with lower values in southern areas and moderate to higher RMSE values in northern and central Europe. The spatial average of BIAS exhibited widespread underestimation, with slight overestimation in the Mediterranean. Introducing a target error from 1.9 mm to 4.2 mm, reduced model's performance as reflected by correlation of 0.91 (-4%) and 0.87 (-8%) with respect to the no error configuration, indicating that the reduction is limited when target error is considered. The values of RMSE increased more significantly to 10.20 mm (+8%) and 12.43 (+32%) suggesting that the absolute accuracy begins to depicts measurable degradation even with smaller error values while the temporal agreement, measured by R, remains preserved. Spatially, both errors keeps higher correlations (R>0.80), consistent error distribution (RMSE < 12 mm) across most of Europe, with slightly less performance in the Eastern European and Scandinavia. Further, the values of BIAS were found stable with mean values significantly low and equal to -0.01 mm and 0.04 mm using 1.9 mm and 4.2 mm error, respectively. The spatial average of BIAS values indicated a tendency towards underestimation over majority of areas with slightly overestimation in the Mediterranean region. This suggests that low values of errors do not induce systematic algorithmic biases while preserving the optimal model operation to simulate precipitation.

At threshold error (19 mm to 42 mm), the model's performance significantly deteriorated with mean R values decreasing to 0.63 (-34%) and 0.32 (-66%) and RMSE increasing to 20.99 mm (+121%) and 24.73 mm (+162%). The spatial pattern indicate poor relationship between simulated precipitation and reference precipitation (R<0.60 & RMSE > 20 mm) in several areas (Northern Scandinavia, scattered areas of Eastern Europe, the Mediterranean region, parts of Alps and surrounding mountainous regions. The results demonstrate that larger errors make it extremely difficult for the model to capture and represent meaningful precipitation patterns with 42 mm error level; reasonable performance were kept with 19 mm error level. Finally, the bias analysis illustrates interesting patterns at threshold error. Mean bias varies between 0.15 mm (19 mm) to 0.04 mm (42 mm). The spatial inspection demonstrate substantial positive bias under 19 mm error which is evident across southern and central Europe, while mix of both positive and negative bias can be evident under 42 mm indicating more complex patterns, suggesting that extreme errors induce spatially variable systematic distortions rather than uniform bias trends.

The model's performance was also evaluated by aggregating the simulated precipitation for a 30 day period, while keeping the same target and threshold errors, i.e. from 1.9 mm to 42 mm; results are shown in Table 1. Starting from the baseline without error, the model maintains good correlation and RMSE under low error conditions of 1.9 mm and 4.2 mm, where correlation decrease up 7% and RMSE increases of 22% (error level of 4.2 mm). As for 15-day precipitation estimation, the performance deteriorates significantly under large error (19 mm to 42 mm). It is also noted that model exhibited enhanced correlation stability under lower error range for the 30 day compared to 15 day




aggregation period. This suggests that longer temporal aggregation helps to preserve the model's ability to capture precipitation dynamics by smoothing out short-term fluctuations.

Table.1 Performance of SM2RAIN algorithm as a proxies of the future NGGM mission forced with ERA5L based TWS at 5 days resolution over Europe for the period 2003-2012 and introducing an error of 1.9 mm to 42 mm (top to bottom in rows) for 30-day aggregation period.

|          | R    | BIAS  | RMSE  |
|----------|------|-------|-------|
|          | (-)  | (mm)  | (mm)  |
| No Error | 0.95 | -0.18 | 14.44 |
| 1.9 mm   | 0.92 | -0.04 | 14.98 |
| 4.2 mm   | 0.88 | 0     | 17.62 |
| 19 mm    | 0.66 | 0.09  | 30.92 |
| 42 mm    | 0.32 | -0.01 | 37.17 |

Differently from R and RMSE, the BIAS remains quite stable in the different configurations. This stability is particularly noteworthy when compared to the spatial patterns observed in the 15-day analysis, where regional bias variations were more pronounced. The 30-day aggregation appears to further stabilize the algorithm's systematic performance, likely due to the temporal averaging effect that reduces the impact of short-term measurement inconsistencies.

#### 3.2 Performance of NGGM\MAGIC mission with respect to GRACE\-FO

The impact of varying temporal resolution (5, 10, 15, 30 days) and error level (from 0 mm to 40 mm) was evaluated at eight randomly selected locations in Italy, Germany, Spain, Greece, Austria, Czechia, Denmark, and Estonia. The results covering the period from 2003 to 2012 for 30-day precipitation estimation are shown in Fig.5. The findings reveal that the temporal resolution has a significant impact on model performance but with varying patterns across error levels. Overall, better temporal resolutions (5-10 days) maintain high R and low RMSE values, even with increasing error, while lower temporal resolutions exhibit sharper drops in correlation as the error grows. However, the rate of degradation in the performance of simulated precipitation varies between countries. Generally, for error values larger than 20 mm, the performance is quite similar for the different temporal resolutions, particularly for the RMSE values. For lower error levels, the differences in the performance for the different temporal resolutions are much larger. This regional representation improves the robustness of the findings by highlighting the model's applicability across diverse climatic conditions.

Figure 5: Evaluation of the NGGM/MAGIC synthetic experiments at four temporal resolutions (5, 10, 15 and 30) with error levels ranging from 0 mm to 40 mm for estimating 30-day precipitation over random locations in Italy, Germany, Spain, Greece, Austria, Czechia, Denmark, Estonia for the period 2003-2012 in terms of correlation coefficient (R) and root mean square error (RMSE).

To obtain more general conclusions on the impact of error level and temporal resolution, another experiment was conducted as presented in Fig.6 were the mean of 100 randomly selected points across Europe were examined similarly to what was done in Figure 5. The analysis revealed a linear decrease in the R-values as a function of error levels and





again more stable performance for error levels larger than 20 mm in terms of RMSE. By setting a threshold value for R to 0.7, above which the performance can be considered satisfactory, for 5-day temporal resolution data the results are satisfactory for error levels lower than 15 mm. For 30-day temporal resolution, the results are not satisfactory even for no error conditions. If we consider the expected error level of NGGM\MAGIC to be lower than 10 mm, we can foresee mean performance better than 0.78 (35 mm) in terms of R (RMSE). The error level of GRACE\-FO was found equal to 25 mm and the temporal resolution is 30 days. The performances is equal to 0.40 (55 mm) in terms of R (RMSE). The expected improvements of NGGM\MAGIC with respect to GRACE are therefore highly significant. Overall, this analysis demonstrated smoother degradation patterns compared to individual country analyses, suggesting that regional challenging areas (like Mediterranean regions) are moderated when averaged across diverse European conditions. The findings also support the 5-10 days temporal resolution which offers optimal performance for precipitation estimation at regional scale.

Figure 6: Evaluation of the NGGM/MAGIC synthetic experiments at four temporal resolutions (5, 10, 15 and 30) with error levels ranging from 0 mm to 40 mm for estimating 30-day precipitation over 100 random points in Europe for the period 2003-2012 in terms of correlation coefficient (R) and root mean square error (RMSE).

## 4. Discussion and Conclusion

Future gravity missions could play a key role in addressing the challenges of interpreting different hydrological components, which remain difficult to capture with the current Gravity Recovery and Climate Experiment (GRACE) and its Follow-On (GRACE-FO) mission. In this study, we aim to address these challenges by providing a comprehensive assessment of upcoming Next-Generation Gravity Mission (NGGM) and Mass-change And Geosciences International Constellation (MAGIC) for precipitation estimation on a European scale. The research applied the SM2RAIN algorithm by inverting the soil water balance equation and for the first time, considered Terrestrial Water Storage (derived from the four soil layers of ERA5L soil moisture as a proxy) for precipitation estimation. The following conclusions can be drawn from the present study:

https://doi.org/10.5194/egusphere-2025-3659 Preprint. Discussion started: 17 October 2025

SM2RAIN algorithm exhibited satisfactory performance to simulate precipitation using TWS data across most of Europe achieving a high correlation of 0.86 and a low RMSE of 13.31 mm for 15-day temporal resolution assessments. The performance of model varied regionally, with stronger correlations found in the Mediterranean region and in the southern Europe, while lower values were observed in the Alpine area in northern Italy, the northeastern states and the coastal regions of Norway.

The verification of SM2RAIN leads to design NGGM/MAGIC configurations which clearly show how temporal resolution and error levels have an impact on the performance for precipitation estimation. The results indicate a high correlation and lower RMSE with shorter temporal aggregation (5-days) compared to longer windows (30-days), even when errors are added.

For error levels lower than 10 mm, the performance deterioration is limited; whereas for error levels larger than 15 mm the performance drop is highly significant. These results highlight the importance of reducing errors threshold in future gravity missions.

The findings of this study provide critical implications for the development and execution of future gravity missions such as NGGM and MAGIC. They strongly advocate the value of high temporal resolution (5-day or better) encompassing low error levels (less than 10 mm) to ensure precise precipitation estimation at both regional and global scale. The results also open new opportunities to estimate precipitation for monitoring hydrological processes from terrestrial water storage (TWS) measurements especially in data-scarce regions. By improving spatio-temporal resolution and accuracy of gravity missions we can enhance our capacity to estimate precipitation globally, with also complementary performance with respect to the use of soil moisture measurements. This important aspect will be investigated in near future investigations.

## **Authors contribution**

LB and MUL designed the study; MUL conducted the analyses; MUL and LB interpreted the analyses; MUL wrote the manuscript draft; MUL, SC, GJ, FL and LB reviewed and edited the manuscript.

#### Data availability

325 All raw and processed data can be available by the corresponding authors upon request.

#### Competing interests

The contact author has declared that none of the authors has any competing interests

## Acknowledgements

The authors acknowledge funding from the Italian Space Agency (ASI) "NGGM/MAGIC, a breakthrough in understanding Earth's dynamics" project (grant agreement No. 2023-22.HH.0.

315

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
