# Peer review of "Beyond GRACE: Evaluating the benefits of NGGM and MAGIC for precipitation estimation over Europe"

_EGUsphere, 2025_

## Referee Comment (RC1)

The manuscript investigates whether precipitation can be retrieved at sub-monthly scales using SM2RAIN by inverting the soil-water balance with terrestrial water storage (TWS) derived from ERA5-Land. It then then assesses performance under synthetic NGGM/MAGIC sampling (5–30 days) and with added noise (≈2–42 mm). Results show good feasibility at 15–30 days aggregation with clean inputs and highlight clear degradation with coarser temporal sampling and higher noise, with NGGM/MAGIC-like settings outperforming a GRACE-FO-like benchmark.

Major comments

- The use of ERA5-Land both to construct TWS (input) and as precipitation reference risks circularity. Validate against independent gauges/radar (ECA&D, national networks) or multi-product references (GPCC, MSWEP, IMERG) over representative regions/seasons.
- The intro motivates higher cadence gravity but doesn't cleanly state why replacing SM with TWS is the key scientific gap. Explicitly articulate why it is expected to improve SM2RAIN (e.g., deeper storage sensitivity).
- Methods resample inputs to 100 km using bilinear interpolation while the resolution of GRACE km and NGGM/MAGIC are lower. Conduct a sensitivity analysis to motivate your selection.
- Nash-Sutcliffe is mentioned but never reported. It is suggested to include.
- Specify calibration design (parameter bounds, train/validation split, spatial cross-validation).
- Add some statistics for the calibrated parameters a, b and Z and discuss how they differentiate from the SM2RAIN other applications.
- While the ERA5 has parameters such as groundwater and snow, the authors calculate "TWS" only by soil moisture. This assumption maybe disqualifies specific areas.
- SM2RAIN neglects runoff and evaporation, which can be reasonable for short events but for 15-30 days, this can be considerable. These parameters should also be considered.
- Clarify why a Gaussian noise model was chosen, how noise is injected (independent per sample? temporally/ spatially correlated?) and precisely how temporal resampling is implemented (averaging/endpoints). Also, how this resample is coupled with the exponential filter step.
- Quantify performance stratified by Köppen–Geiger class and precipitation intensity, not only spatial maps. Relate weak regions (Alps, Baltics, coastal Norway) to process drivers (snowpack, orography, deep storage).

- Compare your model's performance with prior SM2RAIN studies (SM-based) and related inversions. Also, justify the choice of R≥0.7 as "satisfactory" (literature or application-driven).
- Study limitations are missing.
- It is suggested to incorporate discussion with results rather than conclusions to compare your results with prior studies and justify the results.

Editorial comments

- Tighten the abstract starting with motivation and the specific temporal scales actually evaluated (15/30-day), state the calibration/reference clearly and report core numbers with uncertainty. (Minor but in abstract R=0.86 is mentioned as daily, which I think is typo).
- Expand the mission context in the introduction with a concise paragraph of NGGM/MAGIC specs (sampling cadence, effective resolution, target/threshold errors, current program status) and cite appropriately.
- Add a short subsection "Evaluation framework" that upfront defines metrics (R, RMSE, bias, NS), aggregation windows, references used, comparisons performed (feasibility vs synthetic NGGM/MAGIC vs GRACE-FO-like), and the criteria for "satisfactory".
- Split Figure 1 into 1a (domain) and 1b (Köppen–Geiger); improve the caption accordingly and ensure all figures state grid characteristics (masking, number of time steps per pixel).
- Describe how the 100 random points and the eight countries were selected (random seed, spatial/climatic stratification) to ensure it cover everything.
- Clarify the GRACE-FO-like benchmark (≈30-day cadence, ≈25 mm error) as a synthetic reference rather than a run driven by actual GRACE fields and keep these numbers together wherever it appears.
- Prefer quantitative phrasing in conclusions (e.g., deltas in R/RMSE relative to the GRACE-FO-like case under ≤10 mm error and 5-day sampling) and answer the stated research questions directly.
- Consider the use of the word "significantly" evaluating your results. This word is coupled with statistical tests.